# *Prunus Knotted*-like Genes: Genome-Wide Analysis, Transcriptional Response to Cytokinin in Micropropagation, and Rootstock Transformation

**DOI:** 10.3390/ijms24033046

**Published:** 2023-02-03

**Authors:** Giulio Testone, Emilia Caboni, Simone D’Angeli, Maria Maddalena Altamura, Donato Giannino

**Affiliations:** 1Institute for Biological Systems, National Research Council (CNR), Monterotondo, 00015 Rome, Italy; 2Council for Agricultural Research and Analysis of Agricultural Economics, Research Centre of Olive, Fruit and Citrus Crops, 00134 Rome, Italy; 3Department of Environmental Biology, “Sapienza” University of Rome, 00185 Rome, Italy

**Keywords:** *KNOX*, *Prunus* spp., rootstocks, 6-benzyladenine, in vitro shoot multiplication, genetic transformation, gene expression, bioinformatics

## Abstract

*Knotted1*-like homeobox (KNOX) transcription factors are involved in plant development, playing complex roles in aerial organs. As *Prunus* species include important fruit tree crops of Italy, an exhaustive investigation of *KNOX* genes was performed using genomic and RNA-seq meta-analyses. Micropropagation is an essential technology for rootstock multiplication; hence, we investigated *KNOX* transcriptional behavior upon increasing 6-benzylaminopurine (BA) doses and the effects on GF677 propagules. Moreover, gene function in *Prunus* spp. was assessed by Gisela 6 rootstock transformation using fluorescence and peach *KNOX* transgenes. Based on ten *Prunus* spp., *KNOX* proteins fit into I-II-M classes named after Arabidopsis. Gene number, class member distribution, and chromosome positions were maintained, and exceptions supported the diversification of *Prunus* from *Cerasus* subgenera, and that of *Armeniaca* from the other sections within *Prunus*. Cytokinin (CK) cis-elements occurred in peach and almond *KNOX* promoters, suggesting a BA regulatory role in GF677 shoot multiplication as confirmed by *KNOX* expression variation dependent on dose, time, and interaction. The tripled BA concentration exacerbated stress, altered CK perception genes, and modified *KNOX* transcriptions, which are proposed to concur in in vitro anomalies. Finally, Gisela 6 transformation efficiency varied (2.6–0.6%) with the genetic construct, with *35S:GFP* being more stable than *35S:KNOPE1* lines, which showed leaf modification typical of *KNOX* overexpression.

## 1. Introduction

In all eukaryote genomes, TALE transcription factors are typified by a homeodomain (HD) with a three amino acid loop extension between helices I and II. Plant TALEs encompass the KNOTTED-like (KNOX) and BELL-like (BELL) factors that form heterodimers and cooperate in organ development [1]. The Arabidopsis KNOX (KNAT) classification was framed on structural and expression features into class I (HD identity >73% vs. maize *Kn1* and meristem expression; STM, BP/KNAT1, KNAT2, and KNAT6), class II (one intron within the ELK domain and widespread transcription: KNAT3, KNAT4, KNAT5, and KNAT7), and class M (devoid of HD, KNATM), conventionally named KNOXI, KNOXII, and KNOXM. This grouping has been suitable to cluster KNOX from several plant species [2]. Synoptically, class I genes work to maintain meristematic identity and are associated with cell proliferation. Referring to epigeous organs, they have several functions, such as shoot apical meristem formation and development (*STM*) and meristem maintenance (*KNAT6*), leaf shape diversity [3], carpel identity (*KNAT2*), stem inflorescence architecture, and pedicel elongation (*BP*/*KNAT1*), including events of function redundancy [4] and hierarchy (e.g., *BP* can regulate *KNAT2/6* [5]). Some of the class II gene functions have been unveiled in aerial organs; e.g., *KNAT3* and *KNAT7* act together to promote secondary cell wall synthesis in xylem vessels, but antagonistically in that of inter-fascicular fibers [6,7]. *KNAT3*/*4*/*5* double or triple loss of function phenotypes recall those overexpressing *KNOX1* in leaves; hence, class II members are likely to antagonize *KNOXI* functions, while epistasis and reciprocal control have had scarce evidence so far [8]. Recently, class II *KNOX*s were shown to coordinate fruit maturation of arabidopsis and tomato [9]. *KNOX*s cooperate with TFs and hormones. Among the former, BELLs recur as selective partners in regulatory networks; moreover, diverse TFs (AS1/AS2, YAB, SAW, CUC, PRC, JLO) act as *KNOX* upstream regulators [3]. As for hormones, *KNOX*s cope with auxins (AUX), gibberellins (GAs), cytokinins (CKs) [3], brassinosteroids [10], and abscisic acid (ABA) [11], including uni- and bilateral controls. For instance, *BP* mediates GA and CK balance in the shoot apical meristem (SAM) by regulating biosynthesis and catabolic genes, and increased CK levels trigger *BP* transcription.

Computational genome-wide studies on dicot fruit tree KNOX have proliferated [12,13], including *Prunus mume* TALEs [14]; gene function in aerial organs has mainly regarded class I *KNOX* of peach, plum, apple, pear, citrus [13,15,16,17,18]. Overall, class I genes were assessed to take part in the development of SAM, leaf shape, stem, axillary buds, and branching [19,20]. Focusing on *KNAT1*/*BP* orthologs, those of peach and pear (*KNOPE1, PbKNOX1)* share the repressive control of cell expansion and lignin synthesis [13,21]. Less is known about *Prunus* class II genes, though roles in secondary cell wall formation were assessed in forest trees [7]. Some mechanisms of KNOX protein interaction and control are conserved in dicot fruit trees, similarly to Arabidopsis. Briefly, there was evidence for KNOX/BELL interactions in *Prunus* [17]. Moreover, in apple, MdLBD11 (lateral bound domain) regulated plant development and growth by modulating two class I *KNOX*s (highly identical to *STM* and *KNAT1*) by mechanisms analogous to the Arabidopsis counterparts, *AtAS2* and *AtASL1*. Moreover, KNOX–KNOX interactions were proven in citrus shoot development [22]. Referring to hormones, transcriptome-wide surveys showed that *KNOX* transcription is responsive to AUX, methyl jasmonate (MeJA), ethylene, and CK treatments in various organs, highlighting member-specific regulatory mechanisms. In-depth studies in apple evidenced that GA triggers *MdKNOX19* (similar to *KNAT1*) transcription able to regulate *ABI5* expression, responsible for ABA synthesis, forming a regulatory module [23,24]. Finally, the MdBLH/MdKNOX15 interaction (AtBLH1/KNAT2 homologs) represses *MdGA2ox7* that deactivates bioactive GAs in apple tree dwarfism [16].

Micropropagation (MP) is an in vitro technology used by tree nursery industries to trade virus-free, genetically homogeneous rootstocks and cultivars. MP is carried out in an artificial and confined environment that per se is stress-generative (e.g., high nutritive contents, exogenous hormone supply, poor gas exchange, low light intensity), first affecting the plant at the morpho-physiological and epigenetic levels [25] and then affecting its functionality during outdoor adaptation. The main MP phases include explant selection/pretreatment, in vitro culture settlement, shoot multiplication (SM), rooting, and acclimatization of clones [26]. A wide body of literature deals with technical aspects of *Prunus* MP; briefly, start explants are usually shoot tips (apices and leaflets) and stem cuttings that bear leaves and axillary buds [27]. During SM, propagules (syn.: microshoots) become rich in offshoots that are further excised for shoot subculturing. Biologically, shoots develop from leaf axillary buds and from adventitious buds formed ex novo in stems’ inner layers or basal callus (in direct contact with medium). Axillary shoots are expected to be genetically identical to the mother plant, whilst genetic anomalies may occur in adventitious shoots, especially if they derive from callus. Technically, SM media are supplied with hormones, and CK type, concentration, and ratio with other hormones are empirically set to guarantee efficient shooting from axillary buds and preserve propagule health necessary for rooting and acclimation. In *Prunus*, BA, a synthetic CK, is widely used, and excess can cause anomalies in propagules [28,29]. Monitoring responses to stress at the molecular level can reveal the clone’s health status and provide information for solutions to mitigate and/or avoid aberrations. Contextually, the *KNOX*s are known to play roles in aerial organ development by cross talking with hormones and CKs (see above), and the study of their response to CK variation is expected to give insight into causes affecting propagule performance and subsequent acclimatization capacity. Relatedly, the expanding technologies of genome modification in *Prunus* are tightly dependent on MP [30,31]. Specifically, peach (the reckoned *Prunus* model species) is among the most recalcitrant to transformation, mainly due to inefficient regeneration, and thus is difficult to subject to gene function surveys and biotech-based breeding [32], while genetic engineering of several *Prunus* rootstocks has been feasible [33]. Finally, the fine control of TF to improve crop traits has been applied in trees [34] and desired in *Prunus* spp. [35].

This work aimed at providing a genome-wide computational and updated characterization of *KNOX* in *Prunus* fruit trees (*PRUNOX*), focusing on CK regulatory elements, followed by a survey on the *PRUNOX* transcription variation in response to CK during rootstock shoot multiplication. Gene transfer (GFP) was first assayed using the Gisela 6 rootstock system technology. Subsequently, the overexpression of peach *KNOPE1*, which is known to affect leaf shape in trees, was used to explore the gene’s effects in aerial organs and its usefulness as a phenotypic marker of transformation.

## 2. Results

### 2.1. Genome-Wide Characterization of KNOX from Prunus Diploid Fruit Trees

#### 2.1.1. Protein Features

A phylogenetic tree was built (Figure 1A) using KNOX proteins from primary transcripts of ten *Prunus* spp. diploid fruit trees (PRUNOX), and the class I, II, and M members of *Arabidopsis thaliana* were used as an outgroup. Manual curation was mandatory to achieve highly confident coding sequences (Appendix A), and the resulting phyletic groups were named referring to Arabidopsis classes. Among class I, a *PRUNOX* group, here named extra group, fell between the BP-like and *KNAT2/6*-like clades without having specific Arabidopsis counterparts. Overall, PRUNOXI and II maintained distinctive HD traits (identity > 70 and 55–58% vs. that of maize Kn1 HD; Appendix A), class-specific motifs of MEINOX (KNOX1 and KNOX2) and class II-specific residues between the MEINOX and ELK [21,36]. The MEINOX of class M proteins, devoid of HD, was more similar to that of class I than II (30 and 26%). Residue conservation grade was investigated (Figure 1B) in all PRUNOXs, using peach KNOX proteins (KNOPEs) as reference. Briefly, MEINOX, ELK, and HD maintained high identity levels (90–100%) within each type of PRUNOX of the examined species, while a grade of variability mostly occurred in the N- and C-termini (Appendix A). Overall, molecular weights and isoelectric points showed very modest variations among the proteins of a fixed group, except for KNOPE2.1 and KNOPEM2 (Figure 1B, Appendix A). Further analysis was focused on regions within variable stretches (conservation score ≤ 0.75) and bearing amino acid replacements (missense substitution at the DNA level), which turned out to be similar at the physical and chemical levels (Appendix A). Moreover, divergent substitutions were classified and predicted to be tolerated (SIFT score ≥ 0.05), except for one observed within the extra group members.

#### 2.1.2. Genomic Features

As for *Prunus* diploid genomes, 11 *PRUNOX* members were found in all species, except for *P. avium* which had 12 (Appendix A). Moreover, *PRUNOXIs* were more numerous than *PRUNOXIIs* and *PRUNOXMs* and organized in a 6-3-2 module, except for *P. avium* which had a 6-4-2 one. The *PRUNOXs* were scattered on scaffolds/linkage groups (Table 1, Figure 2); hereafter, we provide details for the species (*P. armeniaca*, *P. avium*, *P. dulcis*, *P. mira*, *P. mume*, *P. persica*, *P. salicina*) that were fully assembled into eight chromosomes (Chrs).

As for *PRUNOXI,* the two-copy *STM*-like genes recurred on Chrs 3 and 4; the one-copy *BP*-like members were on Chrs 2 (*Armeniaca* section) and 1 (all the other); the two-copy *KNAT2*/*6*-like genes were on Chrs 2 and 7 (*Armeniaca* sect.) and Chrs 1 and 5 (all the others). The one-copy genes of the class I extra group resided on Chr 6, except that of *P. mume* (Chr 1). As for *PRUNOXII*, the 1–3-copy *KNAT3*/*4*/*5*-like genes were on Chrs 2 and 8 (*Armeniaca* sect.) and on Chrs 1 and 7 (all the others); the one-copy *KNAT7*-like members were on Chr 7 (*Armeniaca* sect.) and Chr 5 (all the others). Finally, the 1–2-copy *KNATM* genes lay on Chr 2 (*Armeniaca* sect.) or on Chrs 1 and 5 (all the others).

Synoptically, the *PRUNOX* intron–exon (IN-EX) organization was mostly conserved as in Arabidopsis, except for the extra group and the class M members (Figure 3), structurally different from the orthologs. Briefly, the *PRUNOXI* consisted of five EXs and four INs, except for *STM*-like genes with four EXs and three INs; moreover, the first IN position recurred in the KNOX1 domain. The *PRUNOXII* mainly harbored six EXs and five INs, except for the *KNAT7*-like group with the five-EX and four-IN organization. The *PRUNOXII* maintained one EN in the ELK, a class II peculiarity. The class M genes had three EXs and two INs, the latter residing respectively in *KNOX1*, similar to the *PRUNOXI*, and in *KNOX2*, as in *PRUNOXII*.

*PRUNOX* genomic organization was further analyzed for colinearity, duplication events, nonsynonymous/synonymous substitutions of colinear *KNOX,* and evolution time analysis using six *Prunus* spp. genomes (Appendix A). Hereafter, we only report major outcomes that classify the *Prunus* multicopy orthologs to Arabidopsis *STM*, *KNAT2*/*6*, *KNAT3*/*4*, and *KNATM* genes as segmental duplications, that is, long stretches of duplicated sequences with high identity. Further manual alignments highlighted significant differences between introns (length/identity) in *STM-*like 1 and *STM-*like 2 gene groups (not shown); hence, transposon-mediated duplication—that is, gene fragments embedded into DNA transposons—might have taken place. Similarly, *KNATM* gene shuffling recalled transduplication. Retroduplication events (retrocopied intronless genes bearing a poly-A tail) were not addressed.

We looked for hormone-related cis elements sited in the *KNOX* promoters (1500 bp upstream the ATG) of peach and almond considering that they are the parents of GF677, a rootstock used for CK assays in this work. *KNOX* genes included a variable number of motifs for ABA, AUX, GAs, and ethylene (Appendix A). Here, manual and bioinformatic tools deepened the search for CK motifs, which recurred in all *PRUNOX* members with similar abundance and positions in both species (Figure 4, Appendix A). The *PRUNOXI* harbored 6–17 binding sites of 5–8 bp. The *PRUNOXII* hosted 7–18 motifs, mostly 5 bp long (>80%), while the *PRUNOXM* contained more than 14 elements.

#### 2.1.3. Transcriptomic Features

We further addressed *Prunus* genome-wide transcription in aerial organs (Figure 5), using publicly available RNA-seq data (Appendix A), and only fruit and leaves recurred in all species. The *PRUNOXI*s (vertical blue bar), regardless of developmental stage, shared low expression (blue z-score values) in fruit and leaves; however, their high transcription levels (orange values) characterized meristem-rich organs such as buds and stems (hosting vegetative/floral meristems and cambium). They were all also active in phloem tissues in tested species. Most of the *PRUNOXII*s (vertical orange bar) showed opposite patterns to *PRUNOXI*s in leaves, though exceptions recurred for *KNOPE4* orthologs, while behaviors in fruits, buds, and phloem varied among members and with species. The trends of class M genes often recalled those of class I.

### 2.2. PRUKNOX Responses to Cytokinin in GF677 Rootstock Microcuttings

#### 2.2.1. Pheno-Histological Features

GF677 rootstock microshoots were grown on media containing 1.7 and 5.1 µM BA. Effects of CK treatment were evidenced by the formation of higher numbers of leaflets and side shoots than controls 10 days post-treatment (dpt), while stem length was unvaried (Figure 6A,F,K, and Table 2). In precocious histological analyses (3 dpt), the shoot apical meristem (SAM) and leaf axillary buds (LABs) of controls did not show anomalies (Figure 6B–D), and stems adjacent to the medium did not produce callus (Figure 6E). In microshoots treated with 1.7 µM BA, SAM and LABs (Figure 6G,H) were similar to controls; some LABs started elongation (Figure 6I), and callus occurred at the stem basis (Figure 6J). At 5.1 µM BA, SAM was unaffected (Figure 6L), though several microshoots bore LABs with normal (Figure 6M) or swollen (Figure 6O) morphology, or buds bulging from stem inner layers (Figure 6N), and callus at the stem basis (not shown).

#### 2.2.2. PRUNOX Expression Patterns

Transcription of peach catalase (*PpCAT*) and CK-responsive (*CKR*) genes was monitored to mark events of oxidative stress and CK perception after BA treatment. *CKRs* were selected among Arabidopsis orthologs with ascertained behavior among the CK oxidases (*PpCKX*), responsive regulators (*PpARR*), and histidine kinases (*PpHK*). A further choice was based on genes acting in aerial organs of *Rosaceae* spp. and harboring CK-responsive motifs in promoters (Appendix A). A statistical analysis of the effects of BA dose (D), time (T), and interaction effect (DxT) was carried out to assess the significance of the influence on transcriptional responses of markers and *PRUNOX* and *BELL* genes (Table 3). As for PRUNOXI, BA concentration had significant effects at different levels on several genes tested. Similarly, time affected the expression of most class I *KNOX* genes (except for *KNOPE6*). DxT occurred for all class I genes except for *STMlike1*.

Hereafter, we briefly describe transcription trends (Figure 7) in time at fixed BA dose followed by some highlights of *KNOX* members’ peculiarities. As for stress markers (Figure 7A,B), *PpCAT1* but not *PpCAT2* was upregulated with time with 1.7 μM BA. At higher BA, both *PpCAT* genes were at least 2-fold upregulated in time course, at higher levels at 24 than 72 h post-treatment (hpt). As for *CKR* markers (Figure 7C,D), *PpCKX6* expression was 2-fold higher at 24 hpt due to 1.7 μM BA, followed by restoration to control levels; *PpARR12* and *PpHK1* did not show significant response. Increasing BA to 5.1 μM caused a 4-fold increase in *PpCKX6* transcription at 24 hpt followed by a level drop later on; the *PpARR12* and *PpHK1* genes were over 2-fold upregulated only at 72 hpt. All markers shared expression increase upon higher BA dose at 72 hpt, depicting a state of intense oxidative stress and modification of CK endogenous metabolism.

Regarding *PRUNOXI* after treatment with 1.7 μM BA (Figure 7E), all members showed higher transcription than controls (except for the unvaried *KNOPE6*). Onwards, all *PRUNOX1* members decreased in expression to control levels, except for *STMlike2*, which was repressed. As for *PRUNOXII* (Figure 7G), *KNOPE3* showed comparable expression to controls with time, *KNOPE4* was only slightly repressed at 24 hpt, and *KNOPE7* maintained upregulation that decreased in time course. Looking at the effects of higher BA concentration on *PRUNOXI* (Figure 7F), transcript upregulation was significant only for *STMlike1*, *KNOPE2*, and *KNOPE2.1* at 24 hpt; onwards, the latter two had mRNA levels similar to control, while *STMlike1* was strongly repressed; the expression of *KNOPE1*, *KNOPE6*, and *STMlike1* increased significantly from 24 to 72 hpt. The *PRUNOXII* (Figure 7H) patterns were quite similar to those at low BA. Synoptically, at low BA, *PRUNOXI* shared the “up and down” regulation pattern with time, while member-specific diversified responses occurred at high BA from 24 to 72 hpt, suggesting that CK concentration alters regulatory mechanisms of *PRUNOXI* more intensely than those of *PRUNOXII*, these latter having a time-unvaried pattern under both BA dosages.

As for *BELL* responses to 1.7 μM BA (Figure 7I), *BEL1*, *BLH2*, *BLH5*, and *BLH6* were repressed at 24 hpt, *BLH1* and *BLH8* were triggered, and *BLH3* was similar to control; afterward, *BLH3* and *BLH8* were significantly repressed while the other members’ statuses were the same as those of controls. In media with higher BA (Figure 7J), *BEL1*, *BLH2*, and *BLH5* were downregulated, while *BLH1* and *BLH8* were upregulated and *BLH*3 and *BLH*6 levels were the same as those of controls at 24 hpt. Subsequently, the *BEL1*, *BLH3*, and *BLH5* mRNA levels were the same as those of controls, while those of *BLH1* and *BLH6* were higher and those of *BLH2* and *BLH8* were lower. Comparing the patterns at different BA dosages, six out of seven members (*BLH6*) shared similar patterns at 24 hpt, while discordant trends occurred for *BLH2*, *BLH3*, and *BLH6* at 72 hpt, suggesting that a CK increase alters *BELL* transcription in a complex way.

#### 2.2.3. Gene Coexpression Analysis

Correlation analysis among gene transcript abundance in response to BA dosage was carried out to assess gene coexpression (Figure 8); hereafter, we only report positive (0.7 ≤ r ≤ 1) or negative (−1 ≤ r ≤ −0.7) correlations with high significance (*p* < 0.01). Overall, class I *STMlike1* and class II *KNOPE4* and -*7* positively correlated with *CKR* genes. As for class I genes, the couples *KNOPE1*/*STMlike2*, *STMlike1*/*KNOPE2*, *STMlike1*/*KNOPE2*.1, and *KNOPE2*/*KNOPE2*.1 showed high coexpression levels, whereas no significant correlation among class II members occurred (for the mentioned thresholds). Class I and II genes only showed positive correlations for the following duets: *STMlike1*/*KNOPE7* and *KNOPE2.1*/*KNOPE7*. Synoptically, most class I *KNOPEs* (*KNOPE1*, *STMlike1* and *KNOPE2*, *KNOPE2* and *KNOPE2.1*) correlated with various *BELLs*, and each member showed positive or negative values depending on counterpart (e.g., *KNOPE2.1* vs. *PpBLH5* and *-6* was negative and positive vs. *PpBLH8*). Finally, after scoring for *KNOX*/*BELL* correlations vs. *CKR*, the *STMlike1*/*KNOPE7*/*PpBLH8* module was positively correlated with *PpCKX6*.

### 2.3. KNOPE1 Overexpression in Gisela 6 Rootstock

Regeneration of leaf explants followed a two-step procedure (details in Section 4) consisting of dark–liquid/light–solid on media RM1 and RM2 (Appendix A); the latter is recommended for Gisela 6. After 6 weeks, regeneration frequency (percentage of explants forming novel shoots) was higher in the RM1/RM1 than in the RM2/RM2 (12.0 vs. 4.0 %), while the average shoot number per regenerating explant was similar (4.3 ± 0.5 vs. 4.5 ± 0.7). Consequently, we opted for RM1/RM1 to proceed with agro-infection (Appendix A); shoots originating from selection media (HYG or PTT) underwent a second selection, and three *35S:GFP* and three *35S:KNOPE1* putative transgenic lines were rescued and named primary transgenic clones (PTCs). Southern blot analysis pointed at independent events of T-DNA multiple insertion in Gisela 6 genomes (Appendix A). Transformation frequency was 2.6% and 0.6%, respectively, for *35S:GFP* and *35S:KNOPE1* events (Appendix A). We subcultured the PTC and recovered six *35S:KNOPE1* plants. All *35S:GFP* clones maintained stable expression over time (Figure 9A–I, Appendix A). Differently (Figure 9S), the molecular analyses showed that the clones harbored the *35S:KNOPE1* transgene in the genome (genomic DNA PCR) but its message was undetected (RT-PCR), suggesting malfunctions (silencing), which we did not further address (e.g., detection of small interfering RNAs). Two plants showed phenotypes with altered margins in leaves (Figure 9K–P and Appendix A), a distinctive trait of plants overexpressing *KNOPE1*/*BP*. Finally, one of the two also showed phenotype reversion later (Figure 9Q).

## 3. Discussion

### 3.1. PRUNOX

The complete *KNOX* catalog was retrieved from the genomes of 11 *Prunus* spp., and fine curation of gene structure was carried out for 35 out of 111 sequences. The phylogenetic tree of deduced proteins confirmed the separation into classes I, II, and M together with respective Arabidopsis orthologs [14]. Here, all 10 *Prunus* spp. had the KNATM branch, as already reported for *Prunus* spp. [14,21], which is a typical class of eudicots [2] and unfound in orchids [40]. A subgroup without Arabidopsis counterparts emerged in PRUNOXI, here named extra group (with peach KNOPE6 as reference), which is not unexpected considering the ample variation of *KNOX* number in dicot fruit trees. Specifically, the main diversity source of KNOPE6-like proteins resided in the N- and C-termini (not shown) as compared to the nearest members (e.g., *KNAT2/6* and *KNAT1*). Moreover, *KNOPE6*-like gene structure differs from all the other *PRUNOXI*s in length and sequence (not position) of introns. Finally, all species maintained one *KNOPE6*-like gene (*P. mume* on Chr 1 and the other nine on Chr 6). These data suggest that *KNOPE6*-like may have specific roles in *Prunus* trees; one may regard fruit development as supported by *KNOPE6* association with a QTL regulating several drupe characteristics [21]. As for protein variability, RNA-seq analyses indicate that coding sequences produce isoforms; for instance, both *P. persica* and *P. mume* harbor 11 *KNOX* genes encoding mRNAs for 17 and 12 proteins, implying splicing events. This was in agreement with a bioinformatic survey on *P. mume* TALEs [14]. For a given PRUNOX, amino acid variations (missense substitutions) among the *Prunus* spp. mainly occurred in N- and C-termini, but they were computed as similar and tolerated in most cases, hence supporting the conserved functions within the kind; however, substitutions potentially affecting functions were scored in KNOPE6 members.

The 10 *Prunus* spp. maintained the *PRUNOX* genomic organization in terms of the total number, class composition, and chromosome positions, with modest exceptions. This supports that the diversification of *KNOX* subfamilies took place [2] before the *Prunus* lineage diversification [41]. Referring to the *Prunus* subgenus, the shared *PRUNOX* colinearity brought out one group inclusive of *Armeniaca* section species (*P. armeniaca* and *P. mume*) and the other consisting of *Persicae* (*P. persica* and *P. mira*), *Amigdalus* (*P. dulcis*), and *Prunus* (*P. salicina*) sections. The separation was consistent with the genus evolutionary history and the high vicinity of *P. mume* and *P. armeniaca* compared to other *Prunus* species [42]. As for the *Cerasus* subgenus, *P. avium* hosted an additional copy of the class II *KNOPE3*-like gene, suggesting a specific role in the species of the deciduous corymbose group [42].

Several works report on *KNOX* responsiveness to CK in fruit tree organs [12,13] despite avoiding a computational search for CK-related motifs. Here, the investigation focused on peach and almond *KNOX* and showed that cis-elements of 5–8 bp occurred in all *PRUNOX* members abundantly and with conserved positions. Several other hormone-responsive cis-elements were scored (ABA, AUX, GAs, and ethylene) consistently with other fruit tree works [12,13,14]. Finally, giving speculation on RNA-seq meta-analysis related to CK, the abundant *PRUNOXI* expression in organs characterized by meristem activity (buds and stems) is consistent with the class I roles in participating in maintaining cells at undifferentiated stages in a context of CK–AUX equilibria necessary for meristem development [43].

### 3.2. PRUNOXI and -II Transcription Response to BA in Micropropagation

BA is routinely applied in MP; at 1.7 μM, increased numbers of leaves and lateral shoots, due to axillary bud activation, were expected [29]. The tripled dosage only raised the side shoot number, but histological sections showed the occurrence of budding from subcortical layers of the stem, of bud morphological abnormalities, and of abundant callus at the surface adjacent to the culture medium. At the molecular level, catalase genes (*CAT*) were upregulated by BA increase, while CK catabolism genes were affected by both concentration and time. These markers highlight a stress status associated with CK degradation as a response to hormone uptake. Indeed, at 72 hpt of BA high dose, the collective induction of *CAT*, CK *oxidases*, and CK *receptors* supported the stress exacerbation. These data are consistent with the concept that BA concentration variation is a multitype stress factor [44] and point at these markers as useful for the rapid monitoring of the physiological and health status of GF677 rootstock propagules.

CK responsive elements in promoters and transcriptional variation at 24 hpt support a rapid BA effect on *PRUNOX*, consistently with *PRUNOXI* triggering 3 h post-BA vascular uptake in stems [14]. Several experiments based on BA administration reveal *KNOX* message variation in long periods post-treatment [16,45]. These data pinpoint that responsiveness depends on supply methods, targeted tissue, and plant species. It is thought that *KNOX* genes respond to endogenous CK variations following BA applications since the BA root uptake in Arabidopsis seedlings did not cause *KNOXI* triggering, which instead occurred after endogenous CK levels were increased [46], and Arabidopsis *KNOX*s do not appear to be genes quickly responsive to BA [47]. In support, work on BA-induced caulogenesis from pine leaf showed that BA affected endogenous CK variation in the long term during which key *KNOXI* genes were responsive [48]. Relatedly, BA in the media can alter the inner CKs balance in GF677 [49]. In our system, *CKR* patterns after 72 hpt of high BA dosage may reflect endogenous CK variation (and hormone ratios) leading to altered *KNOX* profiles. Moreover, two gene regulatory aspects emerged from propagule responses: (a) *PRUNOXI*s are more sensitive to BA dose changes than *PRUNOXII*s (Table 3), in particular *STMlike2*, *KNOPE1*, and *KNOPE2.1*; (b) all *PRUNOXI*s (except for *KNOPE6*) modulate expression in response to BA over time, and only *KNOPE7* does so among *PRUNOXII*s. This would suggest a kind of time-coordinated regulation between classes I and II over time [8] in response to CKs.

The propagule is a multiorgan system and *KNOX*s are multifunctional; therefore, establishing *KNOX*-specific (and/or undesired) effects on developmental changes due to BA increase requires additional experiments, here out of scope. However, considering the class I *KNOX* roles in maintaining cells in an undefined state and/or preventing cell expansion or lignification, the altered regulation could contribute to (a) the miniaturized state, typical of in vitro organs, or (b) tissue disorders that subtend hyperhydric traits [44]. Finally, it is known that *KNOX* and CK generate feedback-loop mechanisms of reciprocal control associated with cellular disorders [17], and these may contribute to the wide number of anomalies described for MP [50].

The tripled BA concentration did not affect stem height but did affect leaf and shoot numbers. This condition led to changes in *PRUNOX*I and *BELL* expression levels (and their correlations) at 72 hpt. However, in some cases, *KNOX*/*BELL* coexpression was preserved. Specifically, the *STMlike1*/*KNOPE7*/*PpBLH8* coexpressed module included genes with orthologs that are involved in caulis development [6,51,52]. Hence, the unvaried gene module behavior may account for the unaffected stem trait. At 72 hpt of 5.1 µM, the increased *PRUNOXI* levels may also reflect the presence of more axillary meristems per propagule or de novo bud formation and shooting. Relatedly, it is known that almond *STM*-like transcription preceded the organization of adventitious meristems [53].

### 3.3. Transformation of Gisela 6 and KNOPE1 Phenotypes

In peach, stable [54] and/or transient [55] gene transfer has improved but remains a laborious, low-efficiency, and cultivar-specific process [54]. Alternatively, to assay technology efficiency and study *KNOPE1* function, we employed the Gisela 6 transformation as used in different labs [56,57,58]. Regeneration efficiency is crucial for transformation; here we found that RM1 medium (regeneration frequency 12%, shoots per explant ca. 4%), based on QL and previously shown to be acceptable for the Montmorency cultivar [57], was also satisfactory for Gisela 6, despite the same authors recommending a specific one based on WPM. We may speculate that material origin and status may subtend the RM1 effect (e.g., different subclones of Gisela 6, hormonal treatments on material supplied by the company, and different in vitro growth conditions regarding light intensity). Finally, indirect organogenesis via callus prevailed, confirming that organogenesis-competent cells lay in wounded leaf mid-ribs [56]. The transformation frequency of Gisela 6 was reported to range from 0.5 to 3% [56,57,58], and here it consistently varied from 0.6 to 2.6%, depending on the vector type and transgene cassette. Two rounds of growth on the selection medium, molecular analyses, and phenotype selection (GFP functionality or leaf margin alteration for *KNOPE1*) were necessary to avoid technical escapes. However, malfunction of *35S:KNOPE1* was detected in clonal lines, together with phenotype reversion. Silencing events associated with transgenesis have been known for a long time [59], and here they were not further investigated (e.g., hypermethylation of transgene promoters, siRNA production due interference caused by the transgene vs. endogenous genes). The *35S:KNOPE1* plants bore leaf margin alteration similarly to *KNOPE1* overexpression in Arabidopsis and several other simple leafed species overexpressing *BP*-like genes [17]. However, considering that *KNOX* overexpression causes pleiotropic and dramatic effects, the fine modulation of *KNOXs* by traditional approaches, such as the guide of time/tissue-specific or inducible promoters, or by novel genome editing strategies is envisaged [60].

## 4. Materials and Methods

### 4.1. Bioinformatic Surveys

The published genomes of 10 diploid *Prunus* species were used (Appendix A). They were *P. persica*, *P. kansuensis*, *P. mira*, *P. ferganesis*, and *P. davidiana* (peaches); *P. armeniaca*; *P. mume* (apricots); *P. avium* (sweet cherry); *P. dulcis* (almond); and *P. salicina* (plum). This study did not include ornamental, hybrid, and polyploid *Prunus* species.

As for genomic analyses, the reference assemblies and gene annotations were derived from the GDR database (www.rosaceae.org (accessed on 27 December 2022)), except for *P. mume* data from the NCBI repository. The keyword search of functional annotations used to identify *PRUNOX* genes was based on the term “KNOTTED” and terms related to KNOX domains (KNOX1, KNOX2, ELK, and HD). A similarity search was achieved using *P. persica* sequences as queries in each genome and using BLASTn (e-value cut-off ≤ 1 × 10^−15^). Gene models and annotations were manually curated (Appendix A). The Gene Structure Display Server 2.0 (http://gsds.gao-lab.org (accessed on 27 December 2022)) was used to draw gene structures and protein domain organization. The MCScanX [61] program was used to perform the colinearity analyses of KNOX proteins within each *Prunus* spp. genome at the chromosome assembly level, and duplication types were classified into segmental, tandem, proximal, and others. Subsequently, colinear *KNOX*s were assessed for nonsynonymous and synonymous substitutions (Ka and Ks values) using PAL2NAL web service (http://www.bork.embl.de/pal2nal (accessed on 27 December 2022)). The formula to calculate the duplication time of colinear KNOX was T = Ks/2λ (with λ = 6.56 × 10^−9^ for dicots) [62]. The *KNOX* colinearity among *Prunus* spp. was determined by comparing chromosome map positions and graphed accordingly. The 1500 bp genomic sequences upstream the start codons of *PRUNOX* were submitted to the PLACE database for plant cis-acting regulatory element analysis. The program did not host CK binding motifs but was useful to detect other hormone-related motifs, which are listed in Appendix A. Additionally, the Regulatory Sequence Analysis Tool 2022 (https://rsat.sb-roscoff.fr (accessed on 27 December 2022)) was run by using the same genomic sequences as above and CK motifs retrieved in [38,39] and reported in Figure 4 and Appendix A.

As for protein analyses, deduced protein sequences were first validated by HMMSCAN (www.ebi.ac.uk/Tools/hmmer/search/hmmscan (accessed on 27 December 2022)) and then used to build a phylogenetic tree in MEGA7 ver. 7 (neighbor-joining method; 1000 bootstrap repetitions). The Scorecons server (www.ebi.ac.uk/thornton-srv/databases/cgi-bin/valdar/scorecons_server.pl (accessed on 27 December 2022)) and SIFT tool (sift.bii.a-star.edu.sg/www/SIFT_related_seqs_submit.html (accessed on 27 December 2022)) produced conservation scores and tolerance substitution scores, respectively. The PRUNOX sequences are in Appendix A.

As for RNA-seq meta-analyses, publicly available transcriptomic data of *Prunus* spp. were mined in the NCBI Sequence Read Archive (SRA; www.ncbi.nlm.nih.gov/sra (accessed on 27 December 2022)) using *Prunus* [orgn:txid = 3754] as query and “RNA”, “Illumina”, “paired”, and “fastq” keywords. The results were visualized in the “SRA Run Selector” tool, and runs from “aerial organs” were selected, giving priority to those with replicates (if any). Overall, 63 paired-end runs were retrieved (Appendix A) using the SRA Toolkit (http://github.com/ncbi/sra-tools (accessed on 27 December 2022)) and used in *PRUNOX* expression meta-analyses. Raw reads were quality-checked and filtered using Trimmomatic v0.39 [63]. High-quality clean reads were aligned to the respective genomes using HISAT2 v2.2.1 [64]; StringTie v2.1.5 [65] was used to assemble transcripts and estimate gene abundance in each sample. *KNOX* expression averages (for all replicates) were converted into z-scores and visualized using gplots package v. 3.1.3 (https://cran.r-project.org/web/packages/gplots/index.html (accessed on 27 December 2022)).

### 4.2. Cytokinin Assays

#### 4.2.1. Micropropagation, Treatments, Sampling, and Histological Analyses

Micropropagated clones of GF677 rootstock (*P. persica* × *P. amygdalus*) were produced by CREA-OFA [29]. Briefly, single buds were grown in vials (Wheaton, IL, USA) on basal medium (BM, 10 mL) made of half-strength MS medium [66], microsalts, vitamins, sucrose (30 g/L), and agar (4 g/L; B&V, Parma, Italy), supplemented with hormones (0.4 mg/L BA, 0.1 mg/L GA3, 0.05 mg/L IAA). After two subcultures (10 days each), shoots were transferred into vessels (Magenta-Sigma, Bologna, Italy) with basal medium (50 mL) supplied with increasing doses of BA (0.0, 1.7, and 5.1 μM), generating shooting media SM 0, 1X, and 3X. The pH was adjusted (5.6) before agar addition, and sterilization followed at 121 °C for 20 min. All chemicals were by Sigma-Aldrich, Milan, Italy unless differently reported. Growth conditions were as follows: 24 ± 2 °C, light/dark—16/8 h, light intensity 37.5 μmol m^−2^ s^−1^ (Osram fluora L58 W/77 lamps, Monaco, Germany). The experimental matrix system consisted of propagules (*n* > 30; 5 p/jar) grown on SM0 (control), 1X, and 3X and replicated to allow sampling after 24 and 72 h and to avoid “open and close” repetitions. Materials for gene expression and histologic analyses derived from the same jars. As for RNA, propagules (*n* ≥ 12) for each treatment and timing were subdivided into three bulks (4 propagules each, named biological replicates), collected in tubes, flash-frozen in liquid nitrogen, and stored at −80 °C.

#### 4.2.2. Morpho-Histological Analyses

Phenotyping was carried out at 10 days post-treatment by photographing 20 propagules per treatment and timing (5 p/jar) and measuring stem height, number of leaves on the main axis, and number of side shoots using ImageJ software (https://imagej.nih (accessed on 27 December 2022)). Student’s *t*-test was used to assess significant differences between treated and untreated samples at a given concentration.

Propagules (*n* ≥ 8) for each treatment and timing were immersed in ethanol (70% *v*/*v*) and stored at 4 °C. The dehydrated samples were embedded in Technovit 7100 (Heraeus Kulzer, Hanau, Germany), longitudinally sectioned at 8 μm with a Microm HM 350 SV microtome (Microm, Neuss, Germany), dried overnight at 40 °C, stained with 1% toluidine blue (*w*/*v*), dried and permanently mounted in Canada Balsam, and observed under light microscopy. Sections were digitally photographed using a Leica DMRB optical microscope equipped with a Leica DC 500 camera.

#### 4.2.3. Gene Expression Analyses

Total RNA was isolated [67], DNAse-treated (RQ1, Promega, Madison, WI, USA), reverse-transcribed (2 µg) by Superscript III (Life Technologies, Carlsbad, CA, USA) at 55 °C using oligo (dT)_20_/random hexamer primer mixture (1:1). The cDNA (100 ng) was amplified in a volume (10 mL) containing Titan HotTaq EvaGreen qPCR Mix 1X (BIOATLAS, San Diego, CA, USA) and 0.3 mM of each primer. Triplicate reactions were performed at the following conditions: 15 min at 95 °C followed by 40 cycles of three-step amplification (30 s at 95 °C, 1 min at 59 °C, and 40 s at 72 °C). Primers are listed in Appendix A. The threshold cycle (Ct) value of each gene was normalized with that of *PpRPII* [68,69] and compared with the Ct of untreated controls using the 2^−ΔΔCt^ method [70].

### 4.3. Gene Transfer into Gisela 6

#### 4.3.1. Plant Materials and In Vitro Culture

We purchased (Vivai Battistini, Cesena, Italy, www.battistinivivai.com (accessed on 27 December 2022)) in vitro clones of Gisela 6 plants. The rootstock is a hybrid derived from *P. cerasus* L. cv. *Schattenmorelle* (4n = 32) × *P. canescens* Bois (2n = 16), ploidy 2n = 3X = 24. Apices from these clones were further subcultured on several media, and the best one—ensuring the culture of vital clones for 40 days—was QBLI (3.3 g/L), which consisted of QL medium [71], vitamins and sucrose (20 g/L), BA (0.5 mg/L), NAA (0.05 mg/L), and agar (6 g/L) at pH 5.2. The clones underwent a 4-week recurrent transfer onto fresh QBLI (habituation); 3-week-old propagules bearing stem and leaves were moved onto McCown’s woody plant medium (WPM, Sigma-Aldrich, Milan, Italy) supplied with NAA (1 mg/L) until rooting was complete (2–3 weeks). Procedures were completed using sterilized (121 °C, 25 min) and film-sealed glass jars placed in growth chambers (16/8 h light/dark, 100 µmol s^−1^ m^−2^ PAR, 25 °C, and 60% HR). In vivo adaptation included clones (5 leaves and well-formed roots) that were transferred from jars to pots (5 × 5 cm, 100% sterilized peat, plastic film coverage) and kept for 2 days in baby rooms (20 °C, RH 80%; 16/8 h light/dark, 80 µmols-1 m-2 PAR). Two weeks later, plantlets (hardened stem, 10 leaves) were moved to larger pots (10 × 10 cm; peat/soil 1:1) in a glasshouse (RH 65–70%, 22–25 °C/spring–summer, 16–18 °C/autumn–winter) under natural light photoperiod and intensity.

#### 4.3.2. Shoot Regeneration

Regeneration tests were as follows: Leaf explants from healthy in vitro clones underwent a 24 h dark pretreatment in liquid medium by gentle shaking and then were transferred to agarized regeneration media (RM) under light. The combinations dark–liquid/light–solid were on RM1/RM1 and RM2/RM2 media; RM2 was recommended for Gisela 6 [57]. RM1 (QL 3.38 g/L+ BA 3 mg/L + NAA 0.5 mg/L + sucrose 4% *v*/*w*) and RM2 (WPM + BA 2 mg/L + IBA 1 mg/L + sucrose 3% *v*/*w*) were at pH 4.8. Explants derived from shoots (length > 1 cm) from clones propagated on QLBI for 3–4 weeks. Leaves (length = 1.5–2 cm) were devoid of petioles and cut with three incisions perpendicular to the main vein (base, middle, and end of lamina). To test regeneration frequency, 50 explants (ca. 5 per Petri dish) were used for each treatment. The RM1/RM1 combination was satisfactory and used for transformation experiments.

#### 4.3.3. Genetic Constructs and Transformation

The *35S:hptII*/*35S:GFP* cassettes of the *pCAMBIA1302* vector, conferring resistance to hygromycin (HYG) and synthetizing the GFP, were used in transformation tests. The *35S:pat*/*35S:KNOPE1* cassettes of the binary vector pBA002, conferring resistance to phosphinothricin (PPT), were previously described [17]. All recombinant plasmids were transferred into the *A. tumefaciens* strain EHA105 by freeze–thaw method.

Single colonies of recombinant *A. tumefaciens* (*At*) were grown (28 °C, 48 h dark) in LB (10 mL) plus antibiotics (nalidixic acid 30 mg/L, spectinomycin or kanamycin 50 mg/L); then, they were collected (2500× *g*, 2 min), suspended in RM1 (25 mL) supplied with acetosyringone (100 µM), and brought to 0.5 value of OD600. The explants were incubated in bacterial suspension (25 mL, 28 °C) for 3 days, and then *At* was removed by washing in RM1 (40 mL, 5 min) plus cefotaxime (500 mg/L) 6 times, followed by 5 washes with RM1 to remove the antibiotic. The explants were dried on sterile filter paper and were laid on solid RM1 plus cefotaxime (250 mg/L) in Petri dishes (6 explants each, 100 × 20 mm dish) for a fortnight at 25 °C. Afterward, the explants were moved onto solid RM1 without cefotaxime and with PPT (20 mg/L) or HYG (10 mg/L). PPT and HYG concentrations were established by previous dosage assays. Putative transgenic clones were screened based on PPT or HYG resistance (absence of chlorosis/necrosis) combined with a phenotypic trait (fringed leaves due to *KNOPE1* or GFP fluorescence), moved to fresh media containing PPT or HYG for further selection for ca. 2 weeks, and finally transferred to rooting medium (without herbicide or antibiotic).

#### 4.3.4. Transgene Analyses

Genome insertion was checked by Southern blot (Appendix A) and PCR analyses. The former was previously detailed for peach [17]; leaf gDNA (30 μg) was EcoRI-digested, fractionated in 0.8% agarose gel, blotted onto a membrane (Hybond N+, Amersham, UK), and hybridized with a probe (25 ng) that was radio-labeled with 5 μL of [32P]-dCTP (45 µL of TE buffer: 10 mM Tris·HCl pH 8.0, 1 mM EDTA) according to Rediprime II kit (Amersham). Overnight hybridization was at 60 °C (buffer NaPi 0.5 M pH 7.2, SDS 5%, EDTA 10 mM), followed by 2 washes (2X and 1X SSC/0.1% SDS, 60 °C, 10 min each) and membrane exposure (12 h, −80 °C) to films (Kodak Biomax, Boston, MA, USA). The *HPTII* probe (ca. 1100 bp) was excised (XhoI, Invitrogen, Waltham, MA, USA) from pCAMBIA1302; the *BAR* probe (ca. 676 bp) was excised (NheI/EcoRI, Buffer M Invitrogen) from *pBA-KNOPE1* plasmid [17]. Fragments were purified (QIAquick gel extraction kit, Qiagen, Valencia, CA, USA) and did not harbor the EcoRI site.

As for PCR, the reaction mixture final conditions in 30 µL were as follows: gDNA (300 ng), primers (0.3 μM), dNTPs (0.2 mM), MgCl_2_ (2.5 mM), 1.25 U Taq polymerase, and 1X Buffer (Dream Taq Fermentas). Cycle conditions were as follows: denaturation start (95 °C/5 min), 35 amplification cycles (95 °C/30 s, 57 °C/30 s, 72 °C/120 s), and final extension (72 °C/5 min). Primers are listed in Appendix A.

In Gisela 6 nontransformed plants, the *KNOPE1*-like messages were not detected or strongly downregulated in leaf blades (devoid of petioles); the use of specific primers allowed the detection of peach *KNOPE1* expression from the transgene *35S:KNOPE1* by RT-PCR in clones previously checked for the transgene in the genome. The cDNA synthesis was described above; the PCR conditions (in 50 µL) were as follows: 200 ng cDNA, 1 µM of each primer, 0.5 mM dNTPs, 2.5U DreamTaq (Fermentas, Waltham, MA, USA), 1X buffer; start cycle 95 °C/5 min, 35 cycles: 95 °C/1 min, 62 °C/30 s, and 72 °C/90 s, final extension: 72 °C/5 min; the products (15 µL) were electrophoresed (0.8% agarose gels). Primers are listed in Appendix A. The fluorescence in *35S:GFP* Gisela 6 was detected using an Eclipse 80i microscope (Nikon, Tokyo, Japan) equipped with FITC filters (B-1E) and NIS-Element BR 2.20 software (Nikon) by directly observing organ tissues of transgenic and control plants.

### 4.4. Statistical Analyses

Data were analyzed by different statistical tests including Student’s *t*-test and two-way ANOVA. Pearson correlations were calculated using the “rcorr” function in the R environment v3.4.3 (Core Team, Vienna, Austria, https://www.R-project.org (accessed on 27 December 2022)).

## Figures and Tables

**Figure 1 ijms-24-03046-f001:**
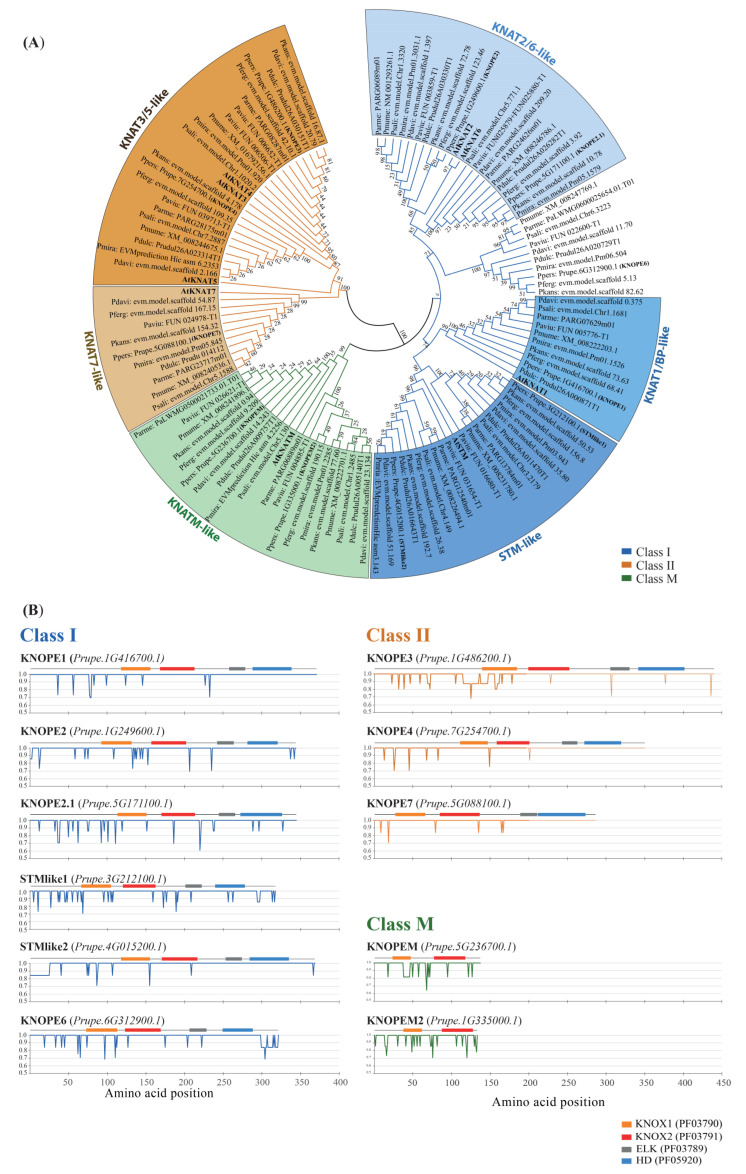
(**A**) Phylogenetic tree of PRUNOX deduced proteins, clustered into classes I, II, and M (blue, orange, and green shades) and named based on the presence of *A. thaliana* (bold) members in the clade. (**B**) Plots of amino acid variability within KNOX subgroups. The peach proteins were schemed (top of each panel) together with key domains (colored boxes) and used as reference for KNOX of other species. Identity variation (percent) of amino acids within *Prunus* spp. proteins (Y-axis) refers to amino acid positions (X-axis). Orange, KNOX1 (PF03790); red, KNOX2 (PF03791); grey, ELK (PF03789); blue, homeodomain (PF05920).

**Figure 2 ijms-24-03046-f002:**
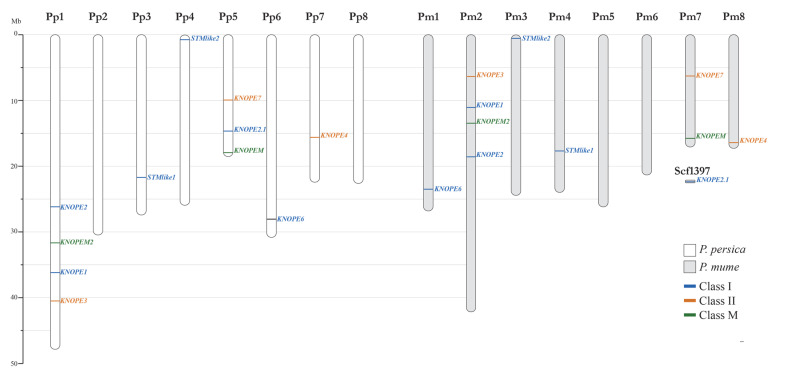
Comparative schematic maps of *KNOX* genome distribution occurring in species of the *Armeniaca* section (e.g., *P. mume*, Pm) vs. that in other *Prunus* spp. (e.g., *P. persica*, Pp). The gene *KNOPE2.1* is located in the unplaced scaffold1397 (Scf1397) in *P. mume* but in Chr 7 in *P. armeniaca*. The scale on the left represents chromosome lengths in megabases (Mb).

**Figure 3 ijms-24-03046-f003:**
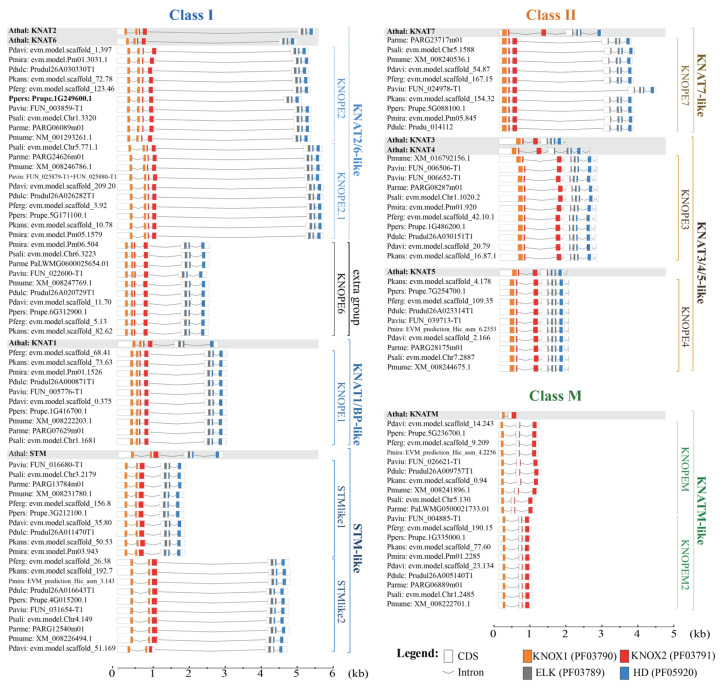
Exon–intron structure and conserved motifs in *PRUNOX* genes. The organization of each class was compared to *A. thaliana* counterparts (grey-shaded). Boxes, coding sequences; gray lines, introns. KNOX conserved motifs are colored.

**Figure 4 ijms-24-03046-f004:**
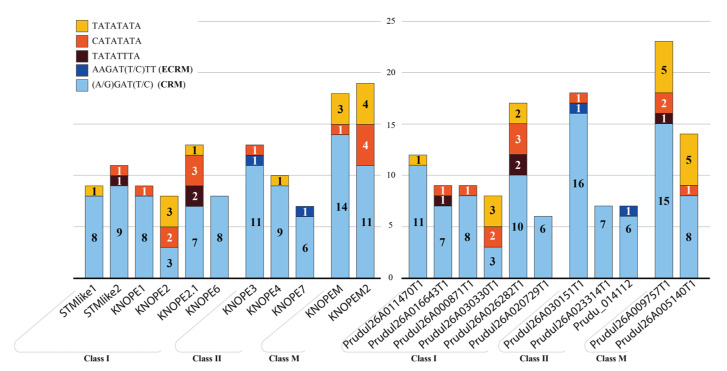
Cytokinin binding motifs residing on *PRUNOX* genes’ promoters of *P. persica* and *P. dulcis* (left and right panels). The score accounts (covering 1.5 kb region before the start codon in Appendix A) refer to the experimentally determined extended (ECRM; blue) and core (CRM; light blue) motifs [38], as well as octameric sequences (yellow/orange/brown) enriched in CK-responsive promoters [39].

**Figure 5 ijms-24-03046-f005:**
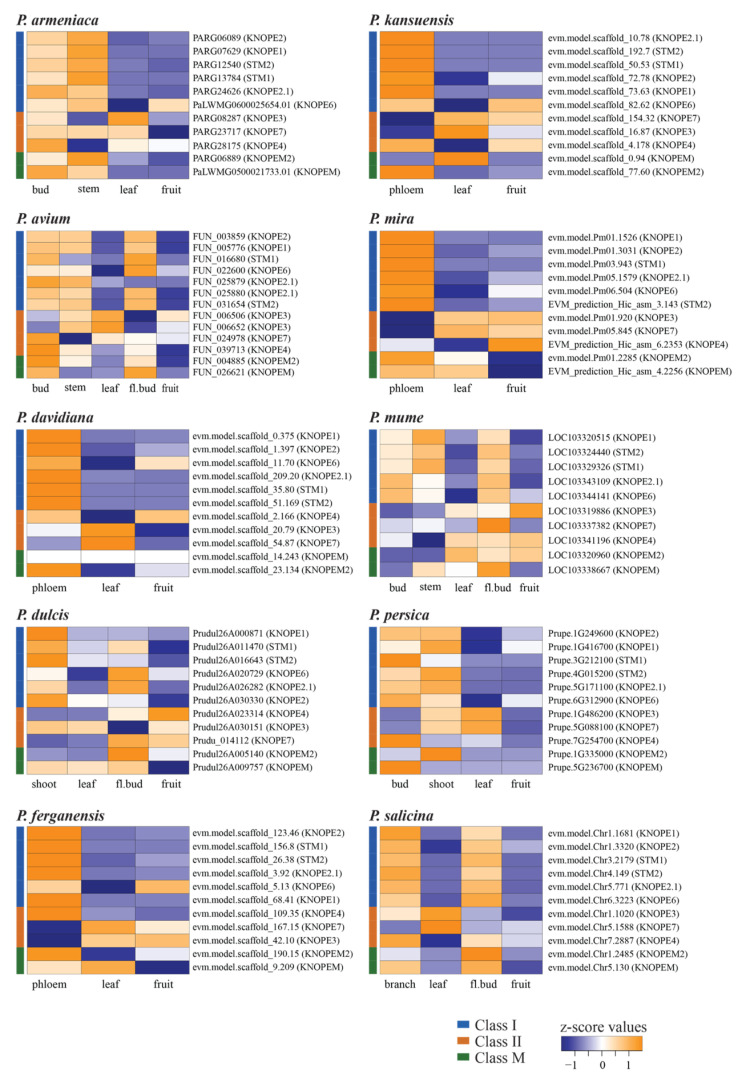
*Prunus* genome-wide transcription analysis of *PRUNOX* genes active in aerial organs. The expression profiles refer to 10 species through heat maps of the z-scores, where orange and blue indicate higher and lower expression, respectively. fl., flower.

**Figure 6 ijms-24-03046-f006:**
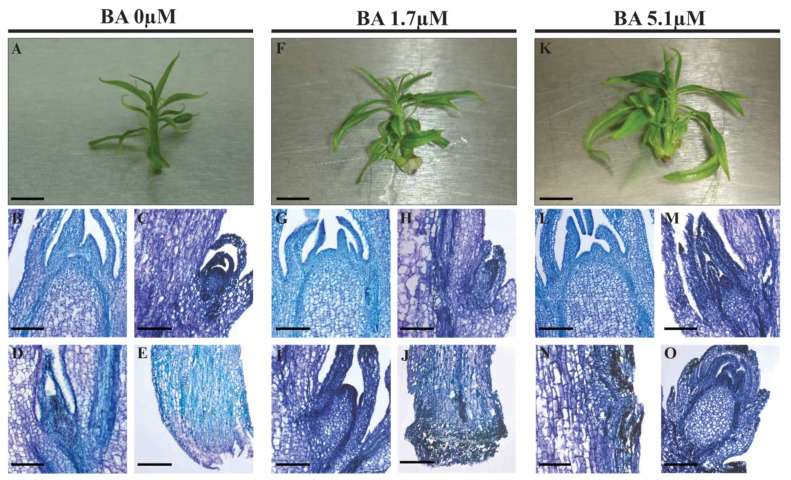
Effects of BA supply on GF677 microcuttings (**A**,**F**,**K**). Explants of the GF677 rootstock were grown on media without BA (**A**–**E**) or containing BA at concentrations of 1.7 (**F**–**J**) and 5.1 µM (**K**–**O**). Ten days post-treatment, the effects of BA dosage in increasing the numbers of side shoots and leaves (compare (**F**,**K**) vs. (**A**)) were visible. (**B**–**E**,**G**–**J**,**L**–**O**) Histological analyses of longitudinal sections at early stages (3 days after treatment). Median section through the dome of control (**B**) and BA-treated apices (**G**,**L**). Leaf axillary buds showed a plethora of phenotypes, including normal (**C**,**D**,**H**,**M**), elongating (**I**), and swollen (**O**). Section of stems portions adjacent to medium (**E**,**J**). Bar sizes: 0.5 cm in (**A**,**F**,**K**); 150 µm in (**D**,**I**,**L**–**N**); 160 µm in (**B**,**G**,**H**,**J**,**L**,**O**); 170 µm in (**C**).

**Figure 7 ijms-24-03046-f007:**
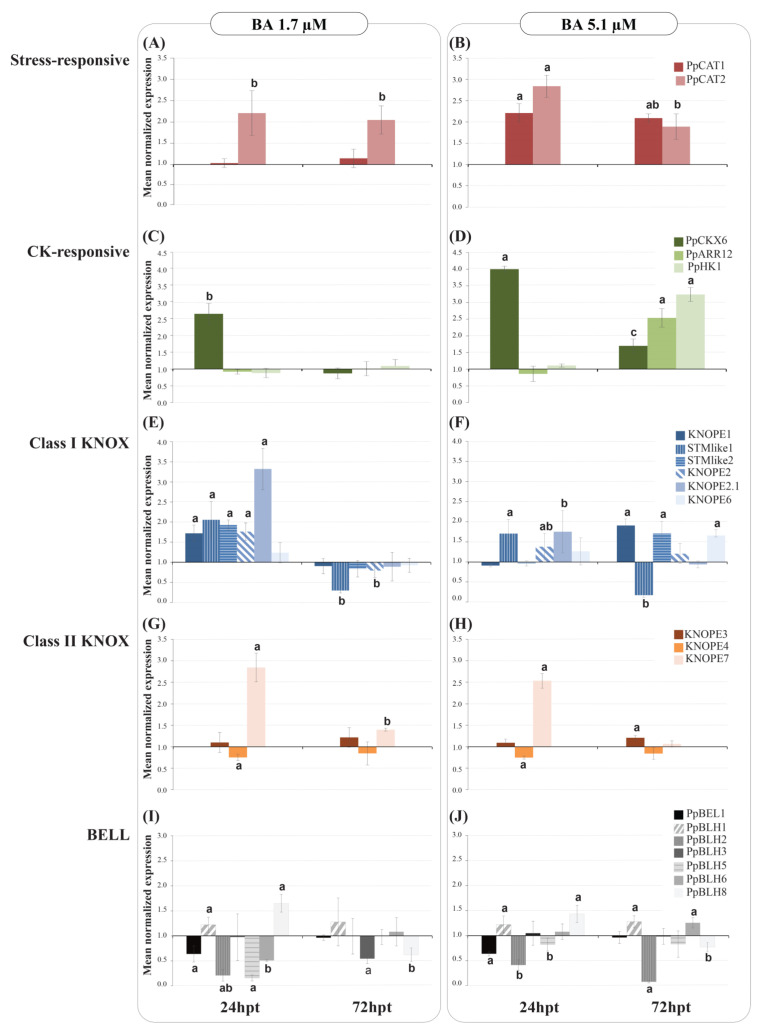
Gene expression analyses of *KNOPE* and marker genes under different BA dosages. Expression levels of stress- (**A**,**B**) and CK-responsive (**C**,**D**) markers and of *KNOPE* (**E**,**H**) and *BELL* (**I**,**J**) genes in response to 1.7 µM (**A**,**C**,**E**,**G**,**I**) and 5.1 µM (**B**,**D**,**F**,**H**,**J**) of BA at 24 and 72 h post-treatment (hpt). Each value represents the mean ± standard error of three replicates. For each gene, different letters mean significant differences (*p* ≤ 0.05).

**Figure 8 ijms-24-03046-f008:**
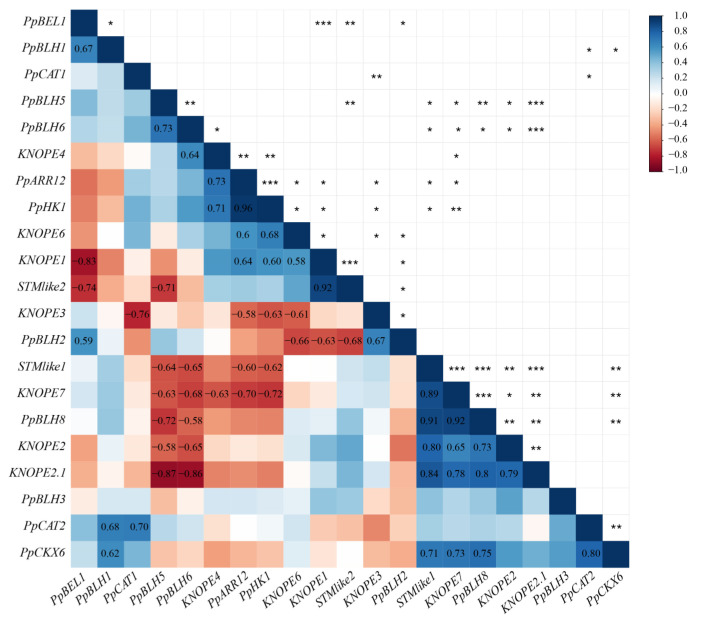
Correlation plot between TALE and marker gene expression levels. Correlogram representing Pearson’s correlation coefficients (r) between TALE (*PRUNOXI*: *STMlike1*, *STMlike2*, *KNOPE1*, *KNOPE2*, *KNOPE2.1*, *KNOPE6*; *PRUNOXII*: *KNOPE3, KNOPE4*, *KNOPE7*; *BELL*: *PpBEL1, PpBLH1, PpBLH2, PpBLH3, PpBLH5, PpBLH6, PpBLH8)* and marker (stress-responsive: *PpCAT1*, *PpCAT2*; CK-responsive: *PpCKX6*, *PpARR12, PpHK1)* expression levels. Heat map is used to indicate the strength of correlation between the variables with ordering determined by hierarchical clustering. Red and blue indicate negative and positive correlations, respectively. Only significant (*p* ≤ 0.05) Pearson’s coefficients were reported in the colored squares. *, **, and *** = significant at *p* ≤ 0.05, 0.01, and 0.001, respectively.

**Figure 9 ijms-24-03046-f009:**
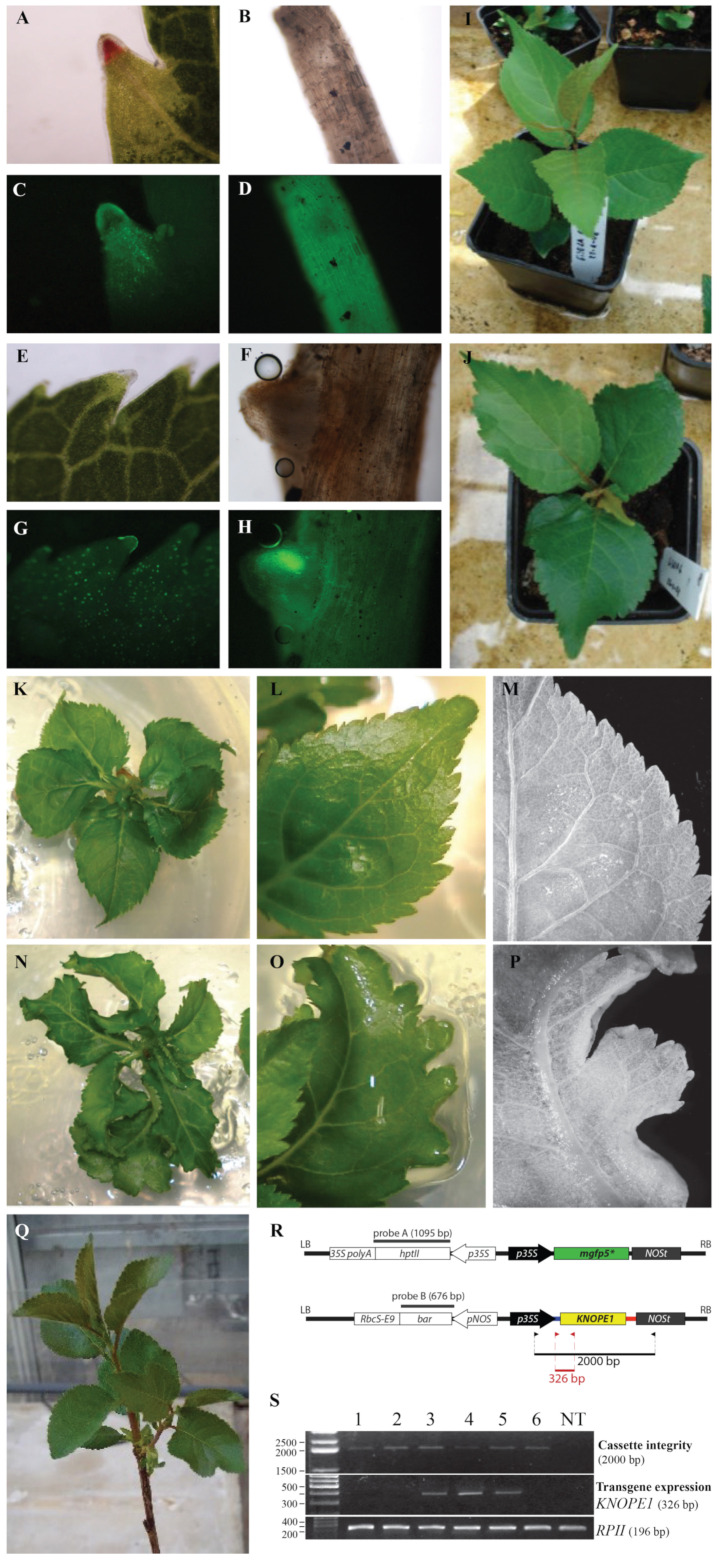
Genetic transformation of Gisela 6. (**A**–**J**) Examples of GFP fluorescence analysis in leaves and roots of Gisela 6 transgenic clones. (**A**–**D**) P1 clone. (**E**–**H**) P3 clone. Analysis of leaf margins (**A**,**C**,**E**,**G**) and lateral roots (**B**,**D**,**F**,**H**) under visible microscopy (**A**,**B**,**E**,**F**) and fluorescence (**C**,**D**,**G**,**H**). (**K**–**Q**) *35S:KNOPE1* phenotypes in Gisela 6. Regenerated clones (**K**,**N**) and details of leaf margins (**L**–**O**) and lamina vasculature (**M**,**P**) in nontransformed (**K**–**M**) and *35S:KNOPE1* (**N**–**P**) lines. (**Q**) *35S:KNOPE1* clone that reverted phenotype with time. (**R**) Vector schemes of *pCAMBIA1302* (above) and *pBA002 + KNOPE1* (below); dark grey bars, probes used in Southern blots which are reported in Appendix A; arrowheads, primers used to check for transgene integrity (black) and expression (red). (**S**) Upper panel, a check for *p35S:KNOPE1:NOSt* cassette integrity by PCR with gDNA. Six clones were rescued and analyzed; on the left, size of bands of DNA ladder in base pairs (bp); on the right, the amplicon size is specified. Mid panel, a check for *KNOPE1* transgene expression in the six clones by RT-PCR using leaf blade RNA. Peach (Chiripa) *KNOPE1*-specific primers fell between the 5’UTR and the first exon (**R**). The constitutive *RPII* expression was assayed to check for correct retrotranscription and usage of equal cDNA amounts. The amplicon sizes are reported.

**Table 1 ijms-24-03046-t001:** Summary of *KNOX* gene members across ten diploid *Prunus* spp. ^1^.

Genus	*Prunus*										Arabidopsis
Subgenus	*Prunus*									*Cerasus*	*-*
Section	*Persicae*					*Armeniaca*		*Amygdalus*	*Prunus*	*-*	*-*
Species ^2^:	*P. dav*	*P. ferg*	*P. kan*	*P. mir*	*P. per*	*P. arm*	*P. mum*	*P. dul*	*P. sal*	*P. avi*	*A. thaliana*
Size (Mb):	237	237	238	243	265	240	280	240	308	338	135
**Class I**											
*STM*-like	2	2	2	2	2	2	2	2	2	2	1
*KNAT1*-like	1	1	1	1	1	1	1	1	1	1	1
*KNAT2*/*6*-like	2	2	2	2	2	2	2	2	2	2	2
extra group	1	1	1	1	1	1	1	1	1	1	0
subtotal	6	6	6	6	6	6	6	6	6	6	4
**Class II**											
*KNAT3*/*4*/*5*-like	2	2	2	2	2	2	2	2	2	3	3
*KNAT7*-like	1	1	1	1	1	1	1	1	1	1	1
subtotal	3	3	3	3	3	3	3	3	3	4	4
**Class M**											
*KNATM*-like	0	1	1	1	1	1	1	1	0	1	1
*KNATM2*-like	2	1	1	1	1	1	1	1	2	1	0
subtotal	**2**	**2**	**2**	**2**	**2**	**2**	**2**	**2**	**2**	**2**	**1**
**Total *KNOX***	**11**	**11**	**11**	**11**	**11**	**11**	**11**	**11**	**11**	**12**	**9**

^1^ Prunus phylogeny and classification as proposed by Zhang et al. [37]. ^2^ Species: *Prunus davidiana*; *P. ferganensis*; *P. kansuensis*; *P. mira*; *P. persica*; *P. armeniaca*; *P. mume*; *P. dulcis*; *P. salicina*; *P. avium*; *Arabidopsis thaliana*.

**Table 2 ijms-24-03046-t002:** Phenotypical traits of GF677 rootstock microcuttings grown on media supplied with different BA dosages.

BA (µM)	Stem Height (cm)	Leaves on Main Axis	Lateral Shoots
0.0	0.9 ± 0.2	12.8 ± 1.5 b	0.0 ± 0.0 c
1.7	1.2 ± 0.3	17.2 ± 3.4 a	2.3 ± 0.1 b
5.1	1.1 ± 0.2	20.3 ± 2.8 a	2.9 ± 0.2 a
Significance	n.s.	***	**

Different letters mean significant differences; ** *p* ≤ 0.01; *** *p* ≤ 0.001; n.s., not significant.

**Table 3 ijms-24-03046-t003:** Significance overview from ANOVA results relative to the gene expression levels in GF677 clones as affected by cytokinin dosage, time, and interaction.

Group	Gene	Dose	Time	DxT
Stress-responsive	*PpCAT1*	**	n.s.	n.s.
	*PpCAT2*	***	n.s.	*
CK-responsive	*PpCKX6*	***	***	n.s.
	*PpARR12*	n.s.	n.s.	***
	*PpHK1*	n.s.	n.s.	***
Class I *KNOX*	*STMlike1*	n.s.	**	n.s.
	*STMlike2*	***	***	***
	*KNOPE1*	***	***	***
	*KNOPE2*	n.s.	**	*
	*KNOPE2.1*	**	***	*
	*KNOPE6*	n.s.	n.s.	*
Class II *KNOX*	*KNOPE3*	n.s.	n.s.	n.s.
	*KNOPE4*	n.s.	n.s.	n.s.
	*KNOPE7*	n.s.	***	n.s.
BELL	*PpBEL1*	**	**	***
	*PpBLH1*	**	n.s.	*
	*PpBLH2*	n.s.	**	**
	*PpBLH3*	n.s.	n.s.	n.s.
	*PpBLH5*	**	***	**
	*PpBLH6*	**	**	n.s.
	*PpBLH8*	n.s.	***	n.s.

n.s., not significant; * *p* ≤ 0.05; ** *p* ≤ 0.01; *** *p* ≤ 0.001.

## Data Availability

Not applicable.

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
