# Peer review of "Prunus Knotted-like Genes: Genome-Wide Analysis, Transcriptional Response to Cytokinin in Micropropagation, and Rootstock Transformation"

_ijms, 2023, doi:10.3390/ijms24033046_

Round 1

Reviewer 1 Report

The manuscript reports a study of Prunus Knotted-like genes based on bioinformatics, BA treatments and gene expression patterns, as well as transgenic materials. The statistical analysis seems globally correct. It's worth noting that only the KNOPE1 was transferred into the rootstock. However, there was no fully explanation in the former contents as to why did the authors choose this gene.

Other suggestions as follows:

(1) Line 207. STM-like genes consisted of 4 EX, which was also different from PRUNOXI.

(2) Line 204-212. It seems class M genes had 3 EX and 2 IN, however there were 2 EX and 1 IN in Arabidopsis. So, “the PRUNOX intron-exon (IN-EX) organisation was as conserved as in Arabidopsis” (line 204-205) is not precise.

(3) It is better to give the reference to the method described in line 218-222.

(4) The results in Table S3 indicated that there were no CK-related cis elements in the promoter regions of P. persica and P. dulcis PRUNOX. It is consistent with the results of KNOX gene family in Orchidaceae (Zhang et al. 2022, doi: 10.3389/fpls.2022.901089). But these were quite different from the results in Figure 4. I suggest adding the promoter sequences to supplementary material and explaining the method in detail.

(5) Line 296-297. It seems only E and J were section of stems portions adjacent to medium. So, “E-J” is an inappropriate description.

(6) Table 2. How many biological replicates?

(7) Section 2.2.2. Which method was used to calculate the gene expressions? And there is also a lack of information on reference genes. I suggest to supplement this information in the Materials and Methods.

(8) Line 333-360. Which were the controls in the Figure 7?

(9) Line 341. KNOPE2 and 2.1 at 24 hptIt is better to describe the genes with the full name. So, replace the “2.1” by “KNOPE2.1”. I recommended to check and modify other similar places in the manuscript.

(10) Line 348. It is hard to understand the two “PRUNOXI”.

(11) Line 414. I suggest listing the genes with classification in the correlogram.

(12) Line 713. What kind of explants were used?

Author Response

We thank the reviewer for critical comments and suggestions that we reckoned as useful to improve the manucript

The manuscript reports a study of Prunus Knotted-like genes based on bioinformatics, BA treatments and gene expression patterns, as well as transgenic materials. The statistical analysis seems globally correct. It's worth noting that only the KNOPE1 was transferred into the rootstock. However, there was no fully explanation in the former contents as to why did the authors choose this gene.

We provided explanation for choosing the KNOPE1 gene in transformation trials. Briefly, KNOPE1 orthologs were previously characterised in other tree species and produced non lethal or extreme (pleiotropic) effects; the typical leaf alteration due to KNOPE1 overexpression was envisaged as marker to better score transgenic lines during the delicate phase of in vitro selection. Morever, 35S:KNOPE1 constructs from our labs were effective in Arabidopsis transgenosis experiments for functional analyses. These concepts were written in the intro and re-called in the Results and Discussions

Lines 135-137: “Subsequently, the overexpression of peach KNOPE1, which is known to affect leaf shape in trees, was used to explore the gene effects in aerial organs and useful as phenotypic marker of transformation”.

Lines: 411-412: “…, a distinctive trait of plants overexpressing KNOPE1/BP.”

Lines: 581-583: “The 35S:KNOPE1 plants bore leaf margin alteration similarly to KNOPE1 overexpression in Arabidopsis and several other simple leafed species overexpressing BP-like genes”

(1) Line 207. STM-like genes consisted of 4 EX, which was also different from PRUNOXI.

We agree with the referee’s observation. The text (lines 207-209) was modified as follows:

“Briefly, the PRUNOXI consisted of 5 EX and 4 IN, except for STM-like genes with 4 EX and 3 IN;”

(2) Line 204-212. It seems class M genes had 3 EX and 2 IN, however there were 2 EX and 1 IN in Arabidopsis. So, “the PRUNOX intron-exon (IN-EX) organisation was as conserved as in Arabidopsis” (line 204-205) is not precise.

We reckon the statement was imprecise. The text (Lines 205-207) was modified as follows:

“the PRUNOX intron-exon (IN-EX) organisation was mostly conserved as in Arabidopsis, except for the extra group and the class M members (Figure 3), structurally different from the orthologues.”

(3) It is better to give the reference to the method described in line 218-222.

We are thankful for this comment ane reckon that furhter explanation is needed. Briefly, we analysed co-linearity and established duplication types of KNOX proteins within each Prunus spp. genome, calculated non-synonymous/synonymous substitutions of collinear KNOX and esteemed evolution times. The whole info is organized in Table S1, while the text of the results just summarizes major features on duplication events for multile copy genes.

In the results the new lines 220-231 were: “PRUNOX genomic organisation was further analysed for collinearity, duplication events, non-synonymous/synonymous substitutions of collinear KNOX and evolution time esteem using six Prunus spp. genomes (Table S1). Hereafter we just report major outcomes that classify the Prunus multi-copy orthologs to Arabidopsis STM, KNAT2/6, KNAT3/4 and KNATM genes as segmental duplications, that is long stretches of duplicated sequences with high identity. Further manual alignments highlighted significant differences between introns (length/identity) in STM-like 1 and STM-like 2 gene groups (not shown); hence transposon mediated duplication – that is gene fragments embedded into DNA transposons- might have taken place. Similarly, KNATM gene shuffling recalled trans-duplication. Retro-duplication events (retro-copied intron-less genes bearing poly-A tail) were not addressed.”.

In the materials and methos the new lines (602-611) were:

“MCScanX [61] program performed the co-linearity analyses of KNOX proteins within each Prunus spp. genome and duplication types were classified into segmental, tandem, proximal and others. Subsequently, collinear KNOX were esteemed for non-synonymous and synonymous substitutions (Ka and Ks values) using PAL2NAL web service (http://www.bork.embl.de/pal2nal). The formula to esteem the duplication time of collinear KNOX was T = Ks/2λ (with λ = 6.56 × 10−9 for dicots [62]. The KNOX colinearity among Prunus spp. was achieved by comparing chromosome map positions and graphed accordingly.”.

(4) The results in Table S3 indicated that there were no CK-related cis elements in the promoter regions of P. persica and P. dulcis PRUNOX. It is consistent with the results of KNOX gene family in Orchidaceae (Zhang et al. 2022, doi: 10.3389/fpls.2022.901089). But these were quite different from the results in Figure 4. I suggest adding the promoter sequences to supplementary material and explaining the method in detail.

A new supplementary table (new Table S4) was compiled that reported the CK motifs from literature - references [38, 39] - and falling in the 1500 bp genomic sequences upstream the start codons of PRUNOX. The whole numbering of the supplementary tables was changed accordingly.

Explanations and clarifications about the analyses of cis-elements in the promoter regions were added in Material and Methods (lines 611-618): “The 1500 bp genomic sequences upstream the start codons of PRUNOX were submitted to the PLACE database for plant cis-acting regulatory elements analysis. The programme did not host CK binding motifs but was useful to detect other hormone related motifs listed in Table S3. Additionally, the Regulatory Sequence Analysis Tool (rsat.sb-roscoff.fr) was run by using the same genomic sequences as above and CK motifs retrieved by literature [38, 39] and reported in Figure 4 and Table S4”.

Legend of Figure 4 was improved (lines 257-261): “Cytokinin binding motifs residing on PRUNOX genes’ promoters of P. persica and P. dulcis (left and right panels). The score accounts (covering 1.5 kb region before the start codon in Table S4) referred to the experimentally-determined extended (ECRM; blue) and core (CRM; light blue) motifs [38], and for octameric sequences (yellow/orange/brown) enriched in CK-responsive promoters [39].”

(5) Line 296-297. It seems only E and J were section of stems portions adjacent to medium. So, “E-J” is an inappropriate description.

The referee’s doubt is correct. We changes the text (lines 295-296) as follows:

“Section of stems portions adjacent to medium (E, J).”.

(6) Table 2. How many biological replicates?

Values in table 2 were measured from 20 propagules per treatment and timing. Details were added in the materials and methods as follows  (lines 664-668):

“Phenotyping was carried out at 10 days post treatment by photographing 20 propagules per treatment and timing (5 p/jar) and measuring stem height, number of leaves on the main axis and of side shoots by ImageJ software (https://imagej.nih). Student’s t-test was used to assess significant differences between treated and untreated samples at a given concentration”.

(7) Section 2.2.2. Which method was used to calculate the gene expressions? And there is also a lack of information on reference genes. I suggest to supplement this information in the Materials and Methods.

We provided further info on relative gene expression quantification and gene normalizers in the Materials and methods section (Lines 686-688).

“The threshold cycle (Ct) value of each gene was normalized with that of PpRPII [68,69] and compared with the Ct of untreated controls using the 2-ΔΔCt method [70].”

(8) Line 333-360. Which were the controls in the Figure 7?

Controls were explants grown on media without BA supply. As for qRT-PCR in Figure 7, gene expression values from controls were used as reference values for relative gene expression quantifications.

(9) Line 341. “KNOPE2 and 2.1 at 24 hpt” It is better to describe the genes with the full name. So, replace the “2.1” by “KNOPE2.1”. I recommended to check and modify other similar places in the manuscript.

We replaced the “2.1” with “KNOPE2.1” along the whole text and figures.

(10) Line 348. It is hard to understand the two “PRUNOXI”.

We corrected the sentence (line 348) as follows 

“…regulatory mechanisms of PRUNOXI more intensely than of PRUNOXII,…”

(11) Line 414. I suggest listing the genes with classification in the correlogram.

We added a remind of gene classification in the Figure 8 legend (line 417-422) as follows:

“Correlogram representing Pearson’s correlation coefficients (r) between TALE (PRUNOXI: STMlike1, STMlike2, KNOPE1, KNOPE2, KNOPE2.1, KNOPE6; PRUNOXII: KNOPE3, KNOPE4, KNOPE7; BELL: PpBEL1, PpBLH1, PpBLH2, PpBLH3, PpBLH5, PpBLH6, PpBLH8) vs markers (stress-responsive: PpCAT1, PpCAT2; CK-responsive: PpCKX6, PpARR12, PpHK1) expression levels.”

(12) Line 713. What kind of explants were used?

We used leaf explants and modified the text (line 712) accordingly.

Reviewer 2 Report

Manuscript titled: Prunus Knotted-like genes: genome-wide analysis, transcriptional response to cytokinin in micropropagation, and rootstock transformation

Authors: Testone et al., 2023

 General Comment. The Manuscript is an excellent and thorough examination of Prunus Knotted like-genes. The work was thorough and accurate and the finding are balanced and well discuss. The reviewer has no main observations on this study.

Minor points: throughout the manuscript ensure that all numbers and units (Including % and Celsius) are spaced out. Check that the consistency of the degree symbol.

Table 1. The heading should be improved as at the subgenus level, correspond the species cerasus too and this is confusing.

Figure 1 a and b, they are both hard to see, and they need to be improved.

Figure 3, very difficult to see and need to be improved.

Figure 5, writing is very pixelated and needs improvement. 

Figure 7. Use bold characters for the y-Axes. Use bold also for significance symbols as currently very difficult to see.

Figure 9. Southern analyses is of poor quality, suggested removal and placement in the supplementary materials.

line 645. change frost by liquid nitrogen to flash frozen in liquid nitrogen

Author Response

We thank the reviewer for appreciating our work and providing suggetions to improve the manuscript

Minor points: throughout the manuscript ensure that all numbers and units (Including % and Celsius) are spaced out. Check that the consistency of the degree symbol.

We checked for proper spacing of numbers and units, and consistency of the degree symbol throughout the article

Table 1. The heading should be improved as at the subgenus level, correspond the species cerasus too and this is confusing.

We modified the table that included complete classification according to Zhang et al [37]. Prunus genus is split into subgenera Prunus and Cerasus, the former including four sections (Persicae, Armeniaca, Amygdalus, Prunus), the latter including the species Prunus avium (and many others here not addressed). Arabidopsis thaliana is used as reference model.

Figure 1 a and b, they are both hard to see, and they need to be improved.

Figure 3, very difficult to see and need to be improved.

Figure 5, writing is very pixelated and needs improvement. 

We produced figures according to the IJMS standards (quality definition greater than 300 dpi); the journal also strongly supports that manuscripts are presented in a word format which requires that figures are embedded in the text. As a consequence, some of them may have lost the original quality and we apologize with the Reviewer for this inconvenient and rendering the comprehension less clear.

In order to enhance the manuscript, figure text fonts were enlarged reducing pixelation effects. In addition, we expect that IJMS will publish on line figures at high definition in specific windows. Should the Referee not be satisfied we will be glad to directly provide her/him with high quality images. Finally, in Figure 5A, F and K, we added the bar sizes and respective values were in the legend: “Bar sizes: 0.5 cm in A, F, K;”…

Figure 7. Use bold characters for the y-Axes. Use bold also for significance symbols as currently very difficult to see.

Thicker trait was used for the y-axis and significance letters were size-increased and bolded.

Figure 9. Southern analyses is of poor quality, suggested removal and placement in the supplementary materials.

We do apologize for the Southern blot image unexpectedly appearing at the panel left side. Original and unprocessed blots are requested by the journal. Hence, we explain the unexpected image due to a cut and paste mistake which was not reckoned during the last control steps.

We modified the Figure 9 by just removing the blots and presenting them into the new Figure S3.

In Figure S3, however, the blots were modified by just enhancing the contrast (minimally altering the original blots) and by adding arrows near the signal bands to support the occurrence of multiple T-DNA copies as stated in the results  (lines: 400-402) and in Table S7 (previously labelled Table S6).

line 645. change frost by liquid nitrogen to flash frozen in liquid nitrogen

We corrected the text as requested (line 661).

Round 2

Reviewer 1 Report

The authors have carefully revised the manuscript. I believe it could be accepted for publication.